# The Evaluation of Activity of Selected Lactic Acid Bacteria for Bioconversion of Milk and Whey from Goat Milk to Release Biomolecules with Antibacterial Activity

**DOI:** 10.3390/molecules28093696

**Published:** 2023-04-25

**Authors:** Agata Biadała, Tomasz Szablewski, Renata Cegielska-Radziejewska, Małgorzata Lasik-Kurdyś, Noranizan Mohd Adzahan

**Affiliations:** 1Department of Food Quality and Safety Management, Faculty of Food Science and Nutrition, Poznan University of Life Sciences, Wojska Polskiego 31, 60-624 Poznan, Poland; 2Department of Food Technology of Plant Origin, Faculty of Food Science and Nutrition, Poznan University of Life Sciences, Wojska Polskiego 31, 60-624 Poznan, Poland; 3Department of Food Technology, Faculty of Food Science and Technology, University Putra Malaysia (UPM), Serdang 43400, Selangor Darul Ehsan, Malaysia

**Keywords:** goat milk, whey, antibacterial activity, lactic acid bacteria

## Abstract

The aim of the study was to assess the antibacterial features of functional macromolecules released during the fermentation of goat milk and whey from goat milk by selected lactic acid bacteria strains that are components of kefir grain microflora. Two milk sources were used: goat milk and whey from goat milk. The lactic acid bacteria (LAB) and indicator microorganisms used were *Lactobacillus plantarum PCM 1386*, *Lactobacillus fermentum PCM 491*, *Lactobacillus rhamnosus PCM 2677*, *Lactobacillus acidophilus PCM 2499*, *Escherichia coli PCM 2793*, *Salmonella enteritidis PCM 2548*, *Micrococcus luteus PCM 525,* and *Proteus mirabilis PCM 1361.* The metabolic activity of LAB was described by the Gompertz model, and the parameters proposed for this experiment were the maximum rate of change of electrical impedance and potential biodegradability. Antibacterial activity was examined using the culture method in a liquid medium, determination of the reduction in indicator microorganisms, and optical density changes. Results show that the selective LAB produced certain active biomolecules with antibacterial activity from whey, a by-product that is sometimes troublesome for goat milk processors to manage. *Lactobacillus acidophilus* is a microorganism that is characterized by the highest metabolic activity in goat milk and whey from goat milk. It has the possibility to produce macromolecules with antibacterial activity.

## 1. Introduction

For decades, human beings have used goats for many purposes [1]. Goat breeding is a traditional activity that was once characteristic of small farms, especially in southern and east-central Europe [2,3]. The main objective of keeping goats is obtaining milk and meat from them, especially in small households with several animals in a herd, mainly because goats are less demanding than other farm animals. The composition and yield of goat milk depend on the breed of goats, environmental conditions, diet, the viability of individual animals, stage of lactation, and climate. Fat content fluctuates the most, while the protein and lactose content in milk is quite stable [4,5]. Changes in the diet of the modern population have resulted in a growing interest in goat milk products as their composition differs from commonly used cow milk. Goat milk is characterized by easier digestibility and lower concentration of αs1-casein, a casein fraction that is responsible for provoking an immune system response. Goat milk also contains more free amino acid content than cow milk. Another advantage of goat milk is about 30% higher content of magnesium and a high content of selenium [4,5,6]. Nowadays, goat milk is mainly used to make cheese. One by-product of the fermentation of goat milk is acid whey. Acid whey is the liquid that remains after the curds have been separated from the milk. It is a rich source of protein and is often used in the production of protein supplements and other food products. Acid whey can also be further processed to extract valuable components such as lactose and minerals or as a raw material for the biotechnological production of bioactive macromolecules. Acid whey can also be used as a highly nutritious animal feed component [2,3]. 

In the dairy industry, membrane processes are commonly used to reduce the number of bacteria in milk. Microfiltration of milk is a process in which milk is passed through membranes with small pores that trap larger particles, such as bacteria, viruses, and blood cells, while allowing smaller particles, such as water, minerals, and proteins, to flow through. One of the products of the microfiltration of milk is whey protein (sweet whey) [3]. 

The use of microfiltration to separate the whey fraction from the casein proteins allows the cheese production process. For the production of cheese, only the fraction with an increased proportion of casein proteins can be used, which significantly increases the production efficiency. The sweet whey fraction can be used as a natural source of nutrients both directly and as a growth medium in biotechnological processes [3,4,5].

Kefir is a fermented milk beverage consumed worldwide. It is comparable to yogurt but differs in several attributes. Kefir is made by a symbiosis of lactic acid bacteria (*Lactobacillus kefiri*, *Lactobacillus fermentum*, *Lactobacillus rhamnosus*, *Lactobacillus brevis*, *Lactobacillus lactis* subsp. *lactis*, *Lactobacillus kefiranofaciens*, *Lactobacillus delbrueckii* subsp. *bulgaricus*, *Leuconostoc mesenteroides*) and yeasts (*Kluyveromyces marxianus*, *Saccharomyces cerevisiae*, *Saccharomyces unisporus*) enclosed in an exopolysaccharide and protein matrix [7,8]. Microorganisms from kefir grain microflora produce a range of antimicrobial compounds, including organic acids, hydrogen peroxide, and bacteriocins, which are effective against a wide range of pathogenic bacteria [2,5,8].

Kefir has documented pro-health properties that allow consumers to maintain optimal health, support the work of the immune system, and help reduce the risk of societal diseases. Kefir is defined as a beverage with antibacterial, antifungal [9,10,11], and antioxidant properties [12,13,14]. There are also data indicating the anticancer properties of kefir [13,15,16]. Literature data showed that kefir grain microflora is a complex of microorganisms characterized by the ability to release and produce compounds with bioactive properties from milk and its fraction (whey) [6,9,16]. 

The ability of microorganisms to grow or multiply in food depends on the food environment. Physical and chemical properties of food and various processing methods play a role in microbial growth. These factors determine not only the microbial growth in food but also the specific metabolic pathways used to generate energy and metabolic by-products [8,9]. 

Factors affecting the growth of microorganisms include temperature, the concentration of hydrogen ions, redox potential, water activity, and hydrostatic pressure [15].

Recent research has shown that LAB in goat milk exhibit high metabolic activity, which is beneficial for both the fermentation process and human health. LAB in goat milk have been found to produce a range of metabolites, including organic acids, exopolysaccharides, and bacteriocins, which have antimicrobial, immunomodulatory, and prebiotic properties [11,12,13,14]. 

The metabolic activity of LAB in goat milk is influenced by various factors, including the composition of the milk, the type and concentration of LAB present, and the conditions of the fermentation process. Optimizing these factors, especially the growth medium (whole milk or whey from goat milk), can result in the production of fermented dairy products with high levels of beneficial metabolites and health-promoting effects [5,6,15].

In the literature, the subject of kefir grain microflora as a starter culture, as well as microorganisms that are part of kefir grain microflora, are described in the context of changes taking place during cow milk fermentation [11,13,14]. Differences in the quantitative and qualitative composition between cow and goat milk give the basis to assume that the activity of individual microorganisms included in the kefir grain and biologically active molecules arising during fermentation will show significant differences. The aim of the study was to assess the activity of individual strains of lactic acid bacteria (which are compounds of kefir grain) to produce biomolecules with antibacterial features from goat milk and compare which source (goat milk or whey from goat milk) is a more valuable medium to produce functional macromolecules.

## 2. Results

### 2.1. The Comparison of Metabolic Activity

Changes in the metabolic activity of selected lactic acid bacteria in analyzed growth media (goat milk and whey from goat milk) followed a second-degree polynomial. The highest degree of matching, expressed by the coefficient R^2^, was characterized by transformations taking place in goat milk (Table 1). The R^2^ match factor was significantly lower for metabolic changes analyzed in whey from goat milk. Whey, as a substrate poorer in casein proteins, is a less attractive substrate for metabolism than whole milk. A more demanding substrate, such as whey, means that microorganisms have to adapt more intensively to the environment, and thus, the course of the lag phase is more irregular.

Based on the calculated parameters, the maximum rate of impedance changes indicating the highest metabolic activity was found in the case of *Lactobacillus acidophilus* in goat milk and for whey in the case of *Lactobacillus plantarum* (Table 2 and Table 3). The highest potential bioconversion capacity was characterized by *Lactobacillus plantarum* and *Lactobacillus acidophilus* in goat milk and *Lactobacillus rhamnosus* in whey.

The mathematical description of curves describing changes in the electrical conductivity of the substrate was used by Paquet et al. [17] during the evaluation of the activity of starter cultures in the cheese production process. The authors described the dynamics of metabolic changes in microorganisms with the parameters of the maximum rate of changes in the conductivity of the environment and the time of its achievement. The rate of changes in the conductivity of the culture medium correlated with the intensity of changes in the acidity of the environment. The authors proposed the use of the parameters of dynamics and the course of changes in environmental conductivity to monitor the activity of starter cultures as an alternative method to measure changes in the acidity of the environment.

### 2.2. The Evaluation of Antibacterial Activity

The evaluation of the inhibition of the growth of indicator microorganisms as a result of metabolic processes performed by lactic acid bacteria from kefir grain microflora indicated that during the fermentation process conducted by *Lactobacillus rhamnosus* metabolites were released that inhibit the growth of *Proteus mirabilis* and *Salmonella enteritidis* (Figure 1 and Figure 2). The bioactive compounds produced during fermentation carried out by *Lactobacillus fermentum* significantly inhibited the growth of *Micrococcus luteus* (Figure 3), while an inhibitory effect was not observed for *Escherichia coli* (Figure 4). The analysis of the curves illustrating the changes in the optical density of goat milk and whey from goat milk undergoing fermentation with selected strains of lactic acid bacteria as a result of inoculation with indicator microorganisms showed that there was no inhibition of *Escherichia coli* growth. During fermentation of selected media with bacteria that are part of the kefir grain microflora, no compounds are formed that inhibit the growth of *Escherichia coli*. The optical density curves are significantly similar to the linear course. 

In all samples analyzed, an inhibition effect for indicator microorganisms began after 5 h of incubation except for *Escherichia coli*, which showed no inhibition effects. The start time for the inhibitory effect was recorded and listed in Table 4. 

Bougherra et al. [18] analyzed the antimicrobial activity of *Lactococcus lactis* subsp. lactis BR16 in bovine milk. Research showed that an inhibition effect was observed after 4 h of fermentation. 

A study on the antibacterial properties of kefir against foodborne pathogens and food spoilage organisms showed inhibitory effects on *Salmonella enteritidis* growth, which was totally inhibited at 36 h and 72 h. However, the same results showed no inhibition effect at 48 h. The reason for these observations can be found in the non-synergistic effect of kefir metabolites. Some metabolites were produced at different times or were degraded during fermentation [19]. 

Lactobacillus bacteria produce primary and secondary metabolites during lactic fermentation. Some of them exhibit antibacterial and antifungal properties, usually directed at a specific group of microorganisms. An example is Lactobacillus casei AST18 which produces the following acids: lactic (93.70 g L^−1^), tartaric (9.59 g L^−1^), lemon (1.29 g L^−1^), acetic (2.42 g L^−1^) and potentially antifungal compounds: cyclo-(Leu-Pro), 2,6-diphenylpiperidine, and 5,10-diethoxy-2,3,7,8-tetrahydro-1H, 6H-dipyrrolo [1,2-a; 1′,2′-d] pyrazine [20]. The above metabolites, acting synergistically (together), are able to inhibit the growth of *Penicillium* sp. Another example of a strain with antimicrobial activity is Lactobacillus curvatus A61, which produces bacteriocins that limit the growth of *Cladosporium* sp., *Fusarium* sp., and *Listeria monocytogenes* [20].

Bougherra et al. [18] identified a new peptide from bovine casein, which shows antibacterial activity against the most deadly strains in the field of food hygiene: *Salmonella enteritidis*, *Escherichia coli*, and *Listeria innocula*.

S-layer proteins on the cell wall in bacteria isolates from kefir have been reported to have antibacterial and antiviral properties [19]. Miao et al. [20] reported that bacteriocin F1 produced by *Lactobacillus paracasei* ssp. tolerans isolated from Tibetan kefir exhibited a wide range of antimicrobial activity against *E. coli*, *Salmonella enterica*, *Shigella dysenteriae*, *Staphylococcus aureus*, *Bacillus thuringiensis*, *Aspergillus flavus*, *Aspergillus niger*, and *Rhizopus nigricans*. The other bacteriocin, Lacticin 3147, which is produced by Lactobacillus lactis DPC3147, showed antibacterial activity against *Yersinia enterocolitica*, *Listeria monocytogenes*, *Salmonella typhimurium*, *Salmonella enteritidis*, and *Shigella flexneri*.

Assessment of inhibition of indicator microorganisms showed that fermented goat milk by *Lactobacillus fermentum* or *Lactobacillus acidophilus* is characterized by the most versatile antibacterial properties against the assessed *Proteus mirabilis*, *Escherichia coli*, *Micrococcus luteus,* and *Salmonella enteritidis* (Table 5). 

The lowest antibacterial properties against indicator microorganisms were found for fermented whey by *Lactobacillus plantarum* and *Lactobacillus fermentum*. 

The quantitative assessment of indicator microorganisms is correlated with the occurrence of inhibition of the assessed bacteria using the plate method. Fermented goat milk by *Lactobacillus fermentum*, *Lactobacillus plantarum,* and *Lactobacillus acidophilus* showed a significantly lower number of colony-forming units compared to the other samples (Table 6). The highest reduction was about 6 logarithmic cycles. 

Fermented whey from goat milk by *Lactobacillus* plantarum and *Lactobacillus fermentum* noted the greatest inhibition (up to 6 logarithmic cycles) of the analyzed indicator microorganisms. The use of the culture method in liquid medium in the experiment allowed the quantitative determination of the growth inhibition effect, which seems to give a better view of the activity of the analyzed lactic acid bacteria than the diffusion method, also used in literature, which is not a quantitative method [21].

Gheziel et al. [22] demonstrated that six *Lactobacillus plantarum* strains isolated from fecal samples expose high antibacterial activity against potential foodborne pathogens *Escherichia coli* and *Staphylococcus aureus*. Likewise, Tarrah et al. [23] reported that *Lactobacillus paracasei* efficiently inhibits *Escherichia coli* and *Listeria monocytogenes* by restraining biofilm formation. Islam et al. [24] found significant growth inhibition of Enterobacter aerogenes by *Lactobacillus plantarum*. 

Saliba et al. [25] evaluated the antibacterial activity of Lactobacillus strains isolated from fermented goat milk. They found significantly higher activity against Gram-positive than Gram-negative bacteria. Inhibition zones against *Escherichia coli* and *Staphylococcus* aureus were found. However, no antibacterial activity was demonstrated against *Listeria monocytogenes*, *Enterococcus faecium,* and *Enterococcus faecalis* [18].

## 3. Materials and Methods

### 3.1. Raw Material

Raw goat milk and whey from goat milk were collected from “Kózka” Organic Farm, Łubowo, Poland.

Goat milk was standardized to 2.5% fat (the initial fat content was 5.4%). The percentage contents of the other components in milk amounted to 3.2% protein and 4.5% lactose.

The whey used in the experiments was a sweet whey obtained from goat milk which was centrifuged, and then the skimmed milk was used in the microfiltration process. Microfiltration was carried out using an Isoflux membrane with a modified filter layer. The temperature at which the process was carried out was 20 °C. The initial pressure was 6 atm. and during the process, it was lowered to 3 atm. Whey was characterized by 0.5% fat, 1.5% protein, and 4.5% lactose content. 

### 3.2. Microbiological Material

The kefir grain microflora and indicator microorganisms used in the experiments were: *Lactobacillus plantarum PCM 1386*, *Lactobacillus fermentum PCM 491*, *Lactobacillus rhamnosus PCM 2677*, *Lactobacillus acidophilus PCM 2499*, *Escherichia coli PCM 2793*, *Salmonella enteritidis PCM 2548*, *Micrococcus luteus PCM 525*, *Proteus mirabilis PCM 1361.*

Microorganisms were obtained from the Polish Collection of Microorganisms at the Institute of Immunology and Experimental Therapy of the Polish Academy of Sciences.

### 3.3. Fermented Milk and Whey Preparation

Goat milk and whey from goat milk obtained after the microfiltration process was inoculated with the analyzed microorganisms listed in Section 3.2. Incubation was run at 37 °C until pH 4.6 was obtained (around 18 h). Samples were cooled to 4 °C after the completion of the fermentation process.

### 3.4. Metabolic Activity Analysis

The metabolic activity of bacteria was assessed using the direct method by recording impedance changes directly in the growth medium using a BacTrac 4100 Automatic Microbial Growth Analyzer [26]. Special tubes with a capacity of 10 mL, equipped with four electrodes, were used to measure the metabolic activity of the bacteria. Each tube was charged with a substrate (9 mL) and 1 mL of inoculum from the test LAB. Changes in the electrical impedance were measured for 24 h at temp. 30 °C. 

Statistical analysis of the results obtained for the assessment of the metabolic activity of microorganisms was carried out using the Curve Export Professional 2.0 program. The experimental curves were described by the Gompertz mathematical model with the following formula:$$y=ae^{-e^{b-cx}}$$

*a*, *b*, *c*—model equation coefficients

*x*—incubation time (h)

*y—*changes in electrical impedance of the growth medium (%).

Based on the Gompertz model, the parameters characterizing the dynamics of changes in the electrical impedance of the substrate were determined:$$Z_{b}=\overset{24}{\underset{0}{\int }}f\left(x\right)dx$$

*I_max_*—maximum rate of change of electrical impedance

*I_max_* = (*a***c*)/*e*

*e* = 2.7183 

*x_I_*—achievement time of *I_max_*

*Z_b_—*potential of bioconversion.

### 3.5. Antimicrobial Properties Testing

#### 3.5.1. Culture Method in Liquid Medium

To investigate if the compounds with antibacterial properties against *Salmonella enteritidis PCM 2548*, *E. coli PCM 2793*, *Micrococcus luteus PCM 525,* and *Proteus mirabilis PCM 1361* were formed, culturing in a liquid medium was performed. The research material was goat milk and whey from goat milk with the addition of lactic acid bacteria *Lactobacillus plantarum PCM 1386*, *Lactobacillus fermentum PCM 491*, *Lactobacillus rhamnosus PCM 2677*, and *Lactobacillus acidophilus PCM 2499*. The tested sample was fermented goat milk or fermented whey from goat milk (2.5 mL), broth (2 mL), and inoculum with indicator microorganisms (0.5 mL). Samples prepared in this way were incubated for 24 h, plated on agar medium, and once again incubated for 24 h at 37 °C. Inoculum of *Salmonella enteritidis*, *Micrococcus luteus,* and *Proteus mirabilis* was prepared at a dilution of 10^6^ cfu/mL, and an inoculum of *E. coli* at a dilution of 10^5^ cfu/mL [27].

#### 3.5.2. Determination of the Reduction in Indicator Microorganisms

The determination was carried out by inoculating plates with a sample (2.5 mL) on selective medium dedicated for indicator microorganisms (inoculum established at 10^8^ cfu/mL) Violet Red Bile Glucose Agar, suitable for microbiology, NutriSelect^®^ Plus, MERCK (Darmstadt, Germany) (VRBG medium), nutrient agar P-0122 BTL, Warszawa, Poland. Then, 2 mL of broth and 0.5 mL of indicator bacteria in the amount of 10^8^ cfu/mL were added to 2.5 mL of milk or whey fermented with the selected LAB. The next step was inoculation for 24 h at 37 °C. Then, after decimal dilutions, the samples were put on selective media and inoculated at 37 °C for 48 h [26].

#### 3.5.3. Optical Density Measurement Using a Bioscreen C

Optical density measurement was performed using Bioscreen C (Oy Growth Curves Ab Ltd., Helsinki, Finland) connected with the software Bioscreen C Pro 2.1.12. The samples (30 μL of inoculum of the test indicator microorganisms, 150 μL of fermented whey or goat milk, and 120 μL of nutrient broth) were incubated at 37 °C for 96 h [27,28]. Optical Density (OD) changes at 600 nm were recorded every 30 min and saved in an Excel spreadsheet coupled with Bioscreen C Pro software.

## 4. Conclusions

Conductive experiments showed that lactic acid bacteria, which are a part of kefir grain microflora, can be released or produced during the fermentation of goat milk and whey from goat milk compounds with antibacterial activity. The combination of microorganisms from kefir grains and goat milk results in a dairy product with higher levels of bioactive molecules and a diverse population of microorganisms that produce a range of antimicrobial compounds. *Lactobacillus fermentum* and *Lactobacillus acidophilus* displayed the highest metabolic activity and the highest potential ability for bioconversion. This was correlated with a high inhibitory effect against the tested indicator microorganisms. Each analyzed lactic acid bacteria showed the ability to metabolize milk or whey from milk to biomolecules, which demonstrates antibacterial activity. In all analyzed samples, the inhibition effect for indicator microorganisms began after 5 h of incubation, except for *Escherichia coli*, which showed no inhibition effects. The lowest antibacterial properties against indicator microorganisms were found for fermented whey by *Lactobacillus plantarum*. Fermented goat milk by *Lactobacillus fermentum*, *Lactobacillus plantarum*, and *Lactobacillus acidophilus* showed the highest reduction in indicator microorganisms, about 6 logarithmic cycles. 

Results indicated that *Lactobacillus acidophilus* is the best microorganism to be a part of a starter culture which, implemented in goat milk manufacturing, will give a product with functional features.

In summary, goat milk is a better source for production molecules with antibacterial activity than whey from goat milk. Whey from goat milk, with the use of an appropriate LAB, can be a substrate for the production of molecules with antimicrobial properties; however, it should be remembered that the efficiency of the process will be lower. Further analysis requires checking the effectiveness of obtaining compounds with antibacterial properties from acid whey obtained after cheese production. The use of such whey would allow for the utilization of the by-product but would not improve the cheese production process itself. This will be achieved by implementing milk after the microfiltration process to production, with an increased ratio of casein proteins, which is key to the formation of the curd.

## Figures and Tables

**Figure 1 molecules-28-03696-f001:**
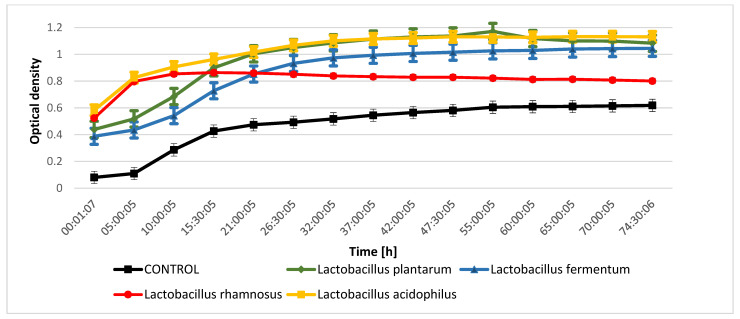
*Salmonella enteritidis* growth changes as a result of metabolic activity of selected LAB.

**Figure 2 molecules-28-03696-f002:**
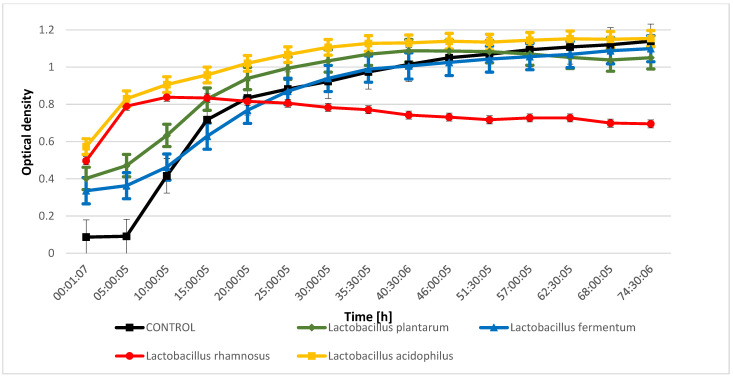
*Proteus mirabilis* growth changes as a result of metabolic activity of selected LAB.

**Figure 3 molecules-28-03696-f003:**
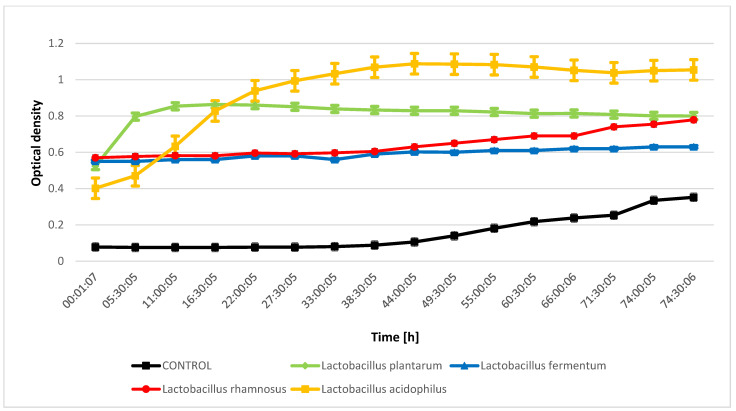
*Micrococcus luteus* growth changes as a result of metabolic activity of selected LAB.

**Figure 4 molecules-28-03696-f004:**
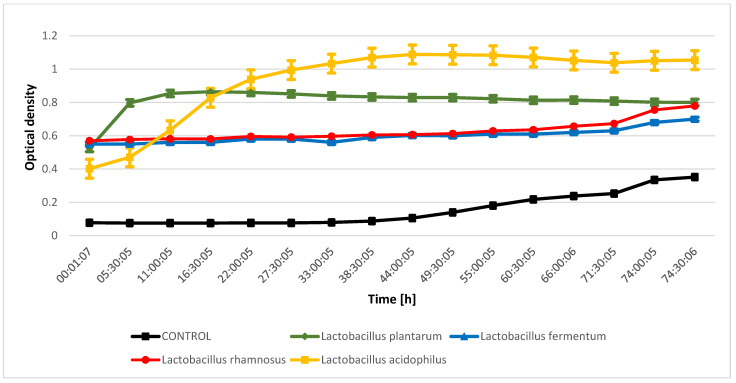
*E. coli* growth changes as a result of metabolic activity of selected LAB.

**Table 1 molecules-28-03696-t001:** Fitting factor R^2^ for the second-degree polynomial describing the change in the metabolic activity of individual bacteria in different growth media.

	*Lactobacillus plantarum*	*Lactobacillus fermentum*	*Lactobacillus rhamnosus*	*Lactobacillus acidophilus*
Goat milk	0.9208 ^B^	0.9335 ^A^	0.9336 ^B^	0.9489 ^B^
Whey from goat milk	0.8850 ^A^	0.9652 ^A^	0.9819 ^A^	0.8514 ^A^

^A,B^—means with different superscripts within same column are significantly different (*p* < 0.05).

**Table 2 molecules-28-03696-t002:** Mathematical parameters of the Gompertz model characterizing the dynamics of changes in electrical impedance in goat milk.

	Gompertz Equation Coefficients				Dynamic Parameters of Impedance Changes		
	a	b	c	r	*I_max_* (ac/e)	*x*_1_ (b/c)	Z_b_ ʃf(*x*)d*x*
*Lactobacillus plantarum*	44.12	5.10	0.48	0.99	6.79 ^C^**^)^	10.63 ^B^	10.06 ^C^
*Lactobacillus fermentum*	31.51	7.38	0.66	0.94	7.65 ^C^	11.18 ^C^	7.02 ^A^
*Lactobacillus rhamnosus*	38.15	4.25	0.37	0.97	5.19 ^B^	11.49 ^C^	8.10 ^B^
*Lactobacillus acidophilus*	45.10	2.36	0.27	0.98	4.48 ^A^	8.74 ^A^	10.64 ^C^

**^)^ different capital letters next to the mean values in the columns mean statistically significant differences at the level of *p* = 0.05.

**Table 3 molecules-28-03696-t003:** Mathematical parameters of the Gompertz model characterizing the dynamics of changes in electrical impedance in goat milk whey.

	Gompertz Equation Coefficients				Dynamic Parameters of Impedance Changes		
	a	B	c	r	*I_max_* (ac/e)	*x*_1_ (b/c)	Z_b_ ʃf(*x*)d*x*
*Lactobacillus plantarum*	28.64	4.44	0.40	0.99	4.21 ^A^**^)^	8.11 ^A^	6.16 ^B^
*Lactobacillus fermentum*	35.19	4.11	0.39	0.96	5.05 ^B^	11.54 ^B^	7.85 ^C^
*Lactobacillus rhamnosus*	45.09	1.92	0.22	0.95	13.65 ^D^	10.73 ^B^	10.48 ^D^
*Lactobacillus acidophilus*	21.02	3.54	0.25	0.98	11.93 ^C^	14.16 ^C^	3.82 ^A^

**^)^ different capital letters next to the mean values in the columns mean statistically significant differences at the level of *p* = 0.05.

**Table 4 molecules-28-03696-t004:** Time (h) after which the start of inhibition of growth of indicator microorganisms in fermented goat milk products was observed (h).

Kefir Grain Microflora	Indicator Microorganisms			
	*Proteus mirabilis*	*E. coli*	*Micrococcus luteus*	*Salmonella enteritidis*
Goat milk				
*Lactobacillus plantarum*	5:38	ND	5:56	6:02
*Lactobacillus fermentum*	5:18	ND	5:31	5:51
*Lactobacillus rhamnosus*	5:02	ND	5:47	5:37
*Lactobacillus acidophilus*	5:25	ND	5:12	5:49
Whey from goat milk				
*Lactobacillus plantarum*	5:58	ND	6:22	6:34
*Lactobacillus fermentum*	6:28	ND	5:29	6:21
*Lactobacillus rhamnosus*	5:32	ND	6:11	5:55
*Lactobacillus acidophilus*	6:35	ND	5:38	5:59

ND—not detected.

**Table 5 molecules-28-03696-t005:** The antimicrobial activity of whey and goat milk fermented by LAB against indicator microorganisms expressed as late growth.

Kefir Grain Microflora	Antimicrobial Activity (Expressed as Late Growth)			
	*Proteus mirabilis*	*E. coli*	*Micrococcus luteus*	*Salmonella enteritidis*
Goat milk				
*Lactobacillus plantarum*	II	I	0	I
*Lactobacillus fermentum*	I	I	0	0
*Lactobacillus rhamnosus*	I	II	I	I
*Lactobacillus acidophilus*	0	I	0	I
Whey from goat milk				
*Lactobacillus plantarum*	II	II	II	II
*Lactobacillus fermentum*	II	II	II	II
*Lactobacillus rhamnosus*	0	II	I	II
*Lactobacillus acidophilus*	II	I	I	I

0—No growth of the microorganisms; I—very slight microorganism growth; II—growth of the microorganism, smaller than the standard.

**Table 6 molecules-28-03696-t006:** The quantitative analysis of antimicrobial activity of whey and goat milk fermented by LAB against indicator microorganisms.

Kefir Grain Microflora	Antimicrobial Activity			
	*Proteus mirabilis*	*E. coli*	*Micrococcus luteus*	*Salmonella nteritidis*
Goat milk				
*Lactobacillus plantarum*	3.1 × 10^3 Bc^	1.2 × 10^5 Ce^	NG ^Aa^	2.1 × 10^3 Bd^
*Lactobacillus fermentum*	1.8 × 10^2 Bb^	2.6 × 10^2 Bb^	1NG ^Aa^	NG ^Aa^
*Lactobacillus rhamnosus*	1.6 × 10^2 Bb^	3.7 × 10^3 Cc^	1.6 × 10^2 Bb^	NG ^Aa^
*Lactobacillus acidophilus*	NG ^A^	4.3 × 10^2 Bb^	1.1 × 10^2 Bb^	1.2 × 10^2 Bc^
Whey from goat milk				
*Lactobacillus plantarum*	NG ^Aa^	NG ^Aa^	2.2 × 10^2 Bb^	1.3 × 10^2 Bc^
*Lactobacillus fermentum*	NG ^Aa^	NG ^Aa^	2.0 × 10^2 Cb^	3.8 × 10 ^Bb^
*Lactobacillus rhamnosus*	1.2 × 10^2 Ab^	1.5 × 10^4 Cd^	1.2 × 10^3 Bc^	4.5 × 10^4 Ce^
*Lactobacillus acidophilus*	1 × 10^3 Cc^	2.6 × 10^2 Bb^	1.3 × 10^3 Cc^	NG ^Aa^

NG*—*no growth; ^a–e^—means with different superscripts within same column are significantly different (*p* < 0.05); ^A*–*C^*—*means with different superscripts within same line are significantly different (*p* < 0.05).

## Data Availability

Not applicable.

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
