# Peer review of "The Evaluation of Activity of Selected Lactic Acid Bacteria for Bioconversion of Milk and Whey from Goat Milk to Release Biomolecules with Antibacterial Activity"

_molecules, 2023, doi:10.3390/molecules28093696_

Round 1
Reviewer 1 Report (Previous Reviewer 1)
Please revise the English laguage since many sentences contain formulations not common for the English language.
Also, the Concluioson part should be revised in ordet to more highlight the obtained results.
Author Response

Reviewer 2 Report (New Reviewer)
The file is attached with comments /corrections

Author Response
Revision Note
Molecules
„The evaluation of activity of selected lactic acid bacteria for bioconversion of milk and whey from goat milk to release biomolecules with antibacterial activity”
Dear Reviewer,
Thank you for your useful comments and suggestions on the language and structure of our manuscript.
We have modified the manuscript accordingly, and detailed corrections are listed below point by point:
- Coorection in file – all correction was implemented in text.
The manuscript has been resubmitted to your journal. We look forward to your positive response.
Dr Agata Biadała
Department of Food Quality and Safety Management
Poznan University of Life Sciences
Wojska Polskiego 31
60-624 Poznan, POLAND
phone: +4861 8466261
email: agata.biadala@up.poznan.pl
Reviewer 3 Report (New Reviewer)
Thank you for submitting the manuscript "The evaluation of activity of selected lactic acid bacteria for bioconversion of milk and whey from goat milk to release biomolecules with antibacterial activity" to Molecules.
Overall, the text needs to be revised as clarity is compromised by the use of short and disjointed sentences. I suggest that authors request a review by a native English speaker.
Line#21: add "and" before Proteus mirabilis.
Abstract: consider reorganizing the abstract so that it is in the sequence of introduction, material and methods, main results and conclusion of the work. As it is, it's confusing.
Line#64: A reference was missing here.
Overall, the introduction needs to be revised. Many claims were made and no references were cited. Authors need to consider that kefir is a well-studied food and much of what was reported in the introduction has been previously studied and is not being verified for the first time by this work.
The axes of the figures that control time need to be improved because, as they are figures full of information, there is a difficulty in understanding what the x axis wants to demonstrate.
Scientific names need to be confirmed as some have typos.
I believe that the introduction should also demonstrate what this research is bringing out differently? What is the innovation of this work in relation to works already reported in the literature? That the fermentation process using kefir can produce metabolites that act with an antimicrobial function is nothing new, as this has been reported for a long time in the literature, in addition to being one of the main appeals for its consumption.
Author Response
Revision Note
Molecules
„The evaluation of activity of selected lactic acid bacteria for bioconversion of milk and whey from goat milk to release biomolecules with antibacterial activity”
Dear Reviewer,
Thank you for your useful comments and suggestions on the language and structure of our manuscript.
We have modified the manuscript accordingly, and detailed corrections are listed below point by point:
- Line#21: add "and" before Proteus mirabilis – done
- Line#64: A reference was missing here. – references was added
- Overall, the introduction needs to be revised. Many claims were made and no references were cited. - the introduction was revised and references were added.
- Abstract: consider reorganizing the abstract so that it is in the sequence of introduction, material and methods, main results and conclusion of the work. As it is, it's confusing. – Abstract is follow the structure: aim of work, material and methods, results and conclusion. According to “Instruction for Authors” it should be without headings.
- The axes of the figures that control time need to be improved because, as they are figures full of -figures were changed, less measured point were described on the axis to clearly improve readability
- Scientific names need to be confirmed - was confirmed and corrected if it was necessary
- I believe that the introduction should also demonstrate what this research is bringing out differently? What is the innovation of this work in relation to works already reported in the literature? That the fermentation process using kefir can produce metabolites that act with an antimicrobial function is nothing new, as this has been reported for a long time in the literature, in addition to being one of the main appeals for its consumption – the introduction was corrected, some paragraphs were added.
The manuscript has been resubmitted to your journal. We look forward to your positive response.
Dr Agata Biadała
Department of Food Quality and Safety Management
Poznan University of Life Sciences
Wojska Polskiego 31
60-624 Poznan, POLAND
phone: +4861 8466261
This manuscript is a resubmission of an earlier submission. The following is a list of the peer review reports and author responses from that submission.
Round 1
Reviewer 1 Report
1.) Results section
apart from typing mistakes (marked yellow) the presentation of Figures needs revision. In the present form it is very hard to distinguish the samples, the used lines and patterns are not distinctive enough
2.) materials&methods section:
a) it is unclear what type of whey was used - acid or sweet, and also the average composition of whey and milk should be enlisted as well
b) it is not clear under which conditions you have preformed incubation of milk and whey , respectively - incubation temperature, time frame?
c) section 3.3.2 needs to be rewritten so that it becomes clear which nutrient media were exactly used for each inspected idnictaor organism, provide information on the used media (Full name, Producer name, town and country of porduction), incubation conditions!
d) section 3.3.3 needs to be revised, provide information on the used equipment (type, producer, country and town...), how was the reading performed after incubation?!
e) section 3.3.4 needs revision, specify the standards, describe in more detail the method, or refer to the method you have used for HPLC analysis.
3.) Conclusions
The conclusions should be rewritten so that they are more specificcaly related to the results of your study. In the current form the conclusions appear too general, without exact data of the investigated relations.

Author Response
Revision Note
Molecules
„Biomolecules with antibacterial properties released from whey and goat milk during fermentation with selected lactic acid bacteria from kefir grain microflora”
Dear Reviewer,
Thank you for your useful comments and suggestions on the language and structure of our manuscript.
We have modified the manuscript accordingly, and detailed corrections are listed below point by point:
Results section
apart from typing mistakes (marked yellow) the presentation of Figures needs revision. In the present form it is very hard to distinguish the samples, the used lines and patterns are not distinctive enough
Typing mistakes have been corrected and colors have been used in the figures to differentiate the data presented in the charts.
materials&methods section:
- a) it is unclear what type of whey was used - acid or sweet, and also the average composition of whey and milk should be enlisted as well
In section “Materials” was clarified that sweet whey was used and added characteristics of the goat's milk and whey used in the experiment.
- b) it is not clear under which conditions you have preformed incubation of milk and whey , respectively - incubation temperature, time frame?
The section 3.3. Fermented milk and whey preparation was added to the manuscript to described condition which were applied to obtained fermented milk and whey.
- c) section 3.3.2 needs to be rewritten so that it becomes clear which nutrient media were exactly used for each inspected idnictaor organism, provide information on the used media (Full name, Producer name, town and country of porduction), incubation conditions!
The full information about media used in the experiments as well as incubation conditions was supplemented in text.
- d) section 3.3.3 needs to be revised, provide information on the used equipment (type, producer, country and town...), how was the reading performed after incubation?!
Section was rewrite and information about equipment and software was added. Also information about the frequency of measurements and the method of its recording.
- e) section 3.3.4 needs revision, specify the standards, describe in more detail the method, or refer to the method you have used for HPLC analysis.
Section was rewrite and supplemented with information on standards, conditions for conducting the analysis and references to the literature on the basis of which the analysis was performed.
Conclusions
The conclusions should be rewritten so that they are more specificcaly related to the results of your study. In the current form the conclusions appear too general, without exact data of the investigated relations.
In conclusion some data and statements from investigated relations ware added for better highlighting the purpose of the research.
The manuscript has been resubmitted to your journal. We look forward to your positive response.
Dr Agata Biadała
Department of Food Quality and Safety Management
Poznan University of Life Sciences
Wojska Polskiego 31
60-624 Poznan, POLAND
phone: +4861 8466261
email: agata.biadala@up.poznan.pl
Reviewer 2 Report
The experimental design is very poor. The discussion should be improved. more results are needed to explain the obtained results (lactic bacteria produce also bacteriocin which have antmicrobial effect). This study is a preliminary results et could not be published as it. The results are not well presented.
Author Response
Revision Note
Molecules
„Biomolecules with antibacterial properties released from whey and goat milk during fermentation with selected lactic acid bacteria from kefir grain microflora”
Dear Reviewer,
Thank you for your useful comments and suggestions.
We have modified the manuscript accordingly.
The experimental design is very poor.
The description of experiment was improved by clarifying individual fragments regarding the course of the experiment, which were unclear and lowered the scientific level of the work.
The discussion should be improved. more results are needed to explain the obtained results (lactic bacteria produce also bacteriocin which have antmicrobial effect). This study is a preliminary results et could not be published as it. The results are not well presented.
Discussion was improved by added more conclusion from literature to discuss result obtained during experiments. Bacteriocins wasn’t analysed but it’s antimicrobial activity was discussed and described in corrected manuscript. The presentation of obtained results was changed (colour charts were implemented) for better presentation.
Conclusion was also rewrite for clearer marking obtained results.
The manuscript was one again corrected by native speaker.
The manuscript has been resubmitted to your journal. We look forward to your positive response.
Dr Agata Biadała
Department of Food Quality and Safety Management
Poznan University of Life Sciences
Wojska Polskiego 31
60-624 Poznan, POLAND
phone: +4861 8466261
email: agata.biadala@up.poznan.pl
Round 2
Reviewer 1 Report
Section 3.1 - Raw material
From your descripton a reader could get the impression that whey was obtained from milk by microfiltration. Is that correct?
To my knowledge, microfiltartion is used for defatting and microbiological stabilization of milk, whey or pother processing liquid. The processing conditions usually applied for MF (pressure, membrane cut off or pore size) can not result in whey separation from milk, so please giev a more clear description of the purpose of the MF applied in your study. I suppose it was used for microbiological stabilization of goat milk. Specify, how did you produce sweet whey, or if you did not produce it than state so.
Results section
Fig. 1 -3: Specify units for Optical density at the y-axis
Descrtiption of figures should be rewritten so that is entirely clear what relations are presented
Reviewer 2 Report
This study is a preliminary results and are not enough to conduct solid conclusions.